# Placenta Previa Complicated with Endometriosis: Contemporary Clinical Management, Molecular Mechanisms, and Future Research Opportunities

**DOI:** 10.3390/biomedicines9111536

**Published:** 2021-10-26

**Authors:** Shinya Matsuzaki, Yoshikazu Nagase, Yutaka Ueda, Mamoru Kakuda, Michihide Maeda, Satoko Matsuzaki, Shoji Kamiura

**Affiliations:** 1Department of Gynecology, Osaka International Cancer Institute, Osaka 541-8567, Japan; maeda.rf@gmail.com (M.M.); kamiura-sh@oici.jp (S.K.); 2Department of Obstetrics and Gynecology, Osaka University Graduate School of Medicine, 2-2 Yamadaoka, Suita, Osaka 565-0871, Japan; ynagase@gyne.med.osaka-u.ac.jp (Y.N.); mamorukakuda@gmail.com (M.K.); 3Osaka General Medical Center, Department of Obstetrics and Gynecology, Osaka 558-8558, Japan; satoko_tsuru@yahoo.co.jp

**Keywords:** placenta previa, endometriosis, pelvic adhesion, cesarean delivery, systematic review

## Abstract

Endometriosis is a common gynecological disease characterized by chronic inflammation, with an estimated prevalence of approximately 5–15% in reproductive-aged women. This study aimed to assess the relationship between placenta previa (PP) and endometriosis. We performed a systematic review of the literature until 30 June 2021, and 24 studies met the inclusion criteria. Using an adjusted pooled analysis, we found that women with endometriosis had a significantly increased rate of PP (adjusted odds ratio (OR) 3.17, 95% confidence interval (CI) 2.58–3.89) compared to those without endometriosis. In an unadjusted analysis, severe endometriosis was associated with an increased prevalence of PP (OR 11.86, 95% CI 4.32–32.57), whereas non-severe endometriosis was not (OR 2.16, 95% CI 0.95–4.89). Notably, one study showed that PP with endometriosis was associated with increased intraoperative bleeding (1.515 mL versus 870 mL, *p* < 0.01) compared to those without endometriosis. Unfortunately, no studies assessed the molecular mechanisms underlying PP in patients with endometriosis. Our findings suggest that there is a strong association between endometriosis and a higher incidence of PP, as well as poor surgical outcomes during cesarean delivery. Therefore, the development of novel therapeutic agents or methods is warranted to prevent PP in women with endometriosis.

## 1. General Overview

Placenta previa (PP) is a well-recognized obstetrical complication of postpartum hemorrhage, and about 50% of pregnant women with PP experience postpartum hemorrhage [1]. Severe postpartum hemorrhage is associated with high maternal morbidity and mortality and leads to approximately 140,000 annual maternal deaths worldwide [2,3,4,5,6]. Moreover, PP is the most significant risk factor for placenta accreta spectrum (PAS). If PP is complicated by PAS, surgical morbidity and mortality drastically increase (e.g., mean blood loss increases from 1200 mL to 3000 mL, hysterectomy rates increase from 3% to 42%, etc.). Recently, endometriosis has been consistently found to be associated with a higher prevalence of PP [7,8] Although the correlation between endometriosis and a higher incidence of PP is robust [9,10,11,12,13], the surgical outcomes in PP patients with endometriosis are unclear.

Endometriosis is an inflammatory benign gynecological condition that involves the presence of dysfunctional endometrial-like stroma and glands frequently with muscular metaplasia and reactive fibrosis outside the uterus [14]. In reproductive-aged women, the estimated prevalence of endometriosis is around 5–15%, and the prevalence of endometriosis is increasing [15,16,17,18]. In fact, in fertile women, the estimated prevalence of endometriosis is approximately 25–50%, although the development of assisted reproductive technology (ART) has led to an increase in pregnancy rates among women with endometriosis [15,16,17,18,19]. Furthermore, the number of PP patients with endometriosis may increase in the future, as ART has increased the rate of successful pregnancies in this population [19,20,21]. In the context of gynecologic surgery, women with endometriosis have been found to have increased rates of ureteral injury and prolonged operative times compared to women without endometriosis [22,23].

For example, as shown in Figure 1, pregnant women with endometriosis often have extrauterine posterior adhesions, making it difficult to exteriorize the uterus. Therefore, PP patients with endometriosis who had a cesarean delivery may have an increased rate of surgical complications compared to PP patients without endometriosis.

We performed a systematic review of computerized databases from their inception until June 2021, synthesizing a narrative review from the findings of previous studies. The primary aim of this review was to focus on the clinical research surrounding PP complicated with endometriosis, as well as studies assessing the molecular characteristics of PP complicated with endometriosis. If no studies were identified in our systematic review regarding the outcome of interest in women with PP complicated with endometriosis, a narrative review about the topic in pregnant women with endometriosis was added.

## 2. Definition of Endometriosis in Previous Studies

We previously performed a systematic review that assessed the effect of endometriosis on the incidence of postpartum hemorrhage and PP [24]. In this systematic review, 19 studies were included and the definition used in each study was reviewed [7,8,11,12,25,26,27,28,29,30,31,32,33,34,35,36,37,38,39]. Of those (*n* = 19), women who had undergone surgery for endometriosis were defined as having endometriosis during pregnancy in 10 studies [7,8,11,27,30,31,32,34,36,37]; four others were defined as having past surgery for endometriosis or clinical diagnosis of endometriosis [12,25,33,35]; four studies used the International Classification of Diseases (ICD)-10 code, and the remaining two studies did not report the criteria they used to define endometriosis. Of the included studies, none used the intraoperative findings of cesarean delivery, histopathological analysis after cesarean delivery, nor clinical diagnosis of endometriosis during pregnancy. Notably, no studies to date have diagnosed endometriosis during pregnancy.

Based on the results of previous studies, and the fact that the accurate diagnosis of endometriosis during pregnancy is difficult, we defined endometriosis during pregnancy in this study as follows: (i) previous surgery for endometriosis; (ii) clinical diagnosis of endometriosis (sonographic findings, pelvic adhesion, uterine posterior adhesion, presence of endometrioma, etc.); (iii) clinical or histopathological diagnosis of endometriosis during cesarean delivery; (iv) classified as endometriosis using an ICD code. In this study, we defined severe endometriosis as those with revised American Society for Reproductive Medicine (rASRM) stage III-IV endometriosis or deep infiltrating endometriosis (DIE).

## 3. Systematic Literature Search

### 3.1. Approach for the Systematic Literature Review

A systematic literature search was conducted to examine the influence of endometriosis on surgical morbidity in women with PP complicated with endometriosis. The search terms were expanded from our previous systematic review to establish the definition of endometriosis defined in this study. The outcomes of interest for this review were as follows: (i) population-based prevalence of endometriosis during pregnancy; (ii) diagnosis of endometriosis in women with PP during pregnancy; (iii) the influence of endometriosis on the incidence of PP; (iv) the influence of endometriosis on the surgical outcomes of PP patients with endometriosis; and (v) molecular research regarding PP complicated with endometriosis.

In accordance with the 2020 edition of Preferred Reporting Items for Systematic Reviews and Meta-Analyses guidelines [40], a systematic search was performed using PubMed, Scopus, and Cochrane Library from their inception to 30 June 2021, using Medical Subject Headings (MeSH) terms (if applicable) and medical terms for the concepts of PP complicated with endometriosis.

### 3.2. Information Sources, Eligibility Criteria, and Search Strategy

Previous articles were screened by their titles, abstracts, and full texts, as previously described with modification [41,42,43,44]. All abstracts were screened by Sa.M. and Sh.M. using the MeSH terms (Appendix A) applied in the PubMed search and Cochrane Library search to identify studies that determined the correlation between endometriosis and PP.

### 3.3. Study Selection

The following inclusion criteria were used: (1) women with endometriosis who met the definition (of endometriosis) in this study; (2) women with pelvic adhesion due to suspected endometriosis (more than half of the patients had a clinical or histopathological diagnosis); (3) original articles that examined the outcome of interest; (4) the incidence of PP complicated with endometriosis was shown; (5) the surgical morbidity of PP complicated with endometriosis was shown; (6) the diagnostic method of PP complicated with endometriosis was shown; and (7) molecular studies that focused on PP complicated with endometriosis.

Studies were excluded according to the following criteria: (1) information was insufficient to identify the outcome of interest; (2) the definition of endometriosis was unclear or did not meet the definition of this study; (3) articles were not written in English; and (4) case reports, case series, letter, conference abstracts, narrative reviews, systematic reviews, and meta-analyses.

### 3.4. Data Extraction

The author (Sh.M.) extracted all data. The variables were recorded as follows: first author’s name, year of study, study location, number of included women, definition of endometriosis, and outcomes of interest. The data to be included in the analysis were double-checked by two authors (L.M. and Sa.M.).

### 3.5. Analysis of Outcome Measures and Assessment of Bias Risk

For the analysis of clinical information, we aimed to assess the effect of endometriosis on the surgical outcome of PP. The co-primary objective of this study was to examine the effect of endometriosis on the incidence of PP and surgical outcomes during cesarean delivery in women with PP. Moreover, we examined the effect of severe endometriosis, such as rASRM stage III-IV endometriosis and DIE, on the prevalence of PP. The second objective of this study was to review the diagnosis of women with PP complicated with endometriosis. Further study outcomes included the molecular characteristics of PP complicated with endometriosis.

### 3.6. Meta-Analysis Plan

Using the eligible data, the risks of the incidence of PP were computed using the 95% confidence intervals (95% CI) of the related values to calculate the odds ratios (OR) for the incidence of PP. The heterogeneity across the studies was examined using *I*^2^ statistics to calculate the proportion of total variation. Per version 6.0 of the *Cochrane Handbook for Systematic Reviews of Interventions*, heterogeneity was examined based on the *I^2^* value. The severity of heterogeneity was defined as the following modifications: low heterogeneity (0–30%); moderate heterogeneity (30–60%); substantial heterogeneity (50–90%); and considerable heterogeneity (75–100%) [45].

A meta-analysis was also performed, and all images were produced using the RevMan ver. 5.4.1 software (Cochrane Collaboration, Copenhagen, Denmark). During the pooled analysis, a fixed-effect analysis was applied to the studies with low heterogeneity, and a random-effect analysis was applied to the studies with moderate to considerable heterogeneity.

### 3.7. Statistical Analyses

Differences in patients’ demographics between the two groups were analyzed using the chi-square test or Fisher’s exact test, as appropriate [46]. All statistical analyses were based on two-sided hypotheses, and the threshold for statistical significance was set at a *p*-value of less than 0.05. The Statistical Package for Social Sciences (IBM SPSS, version 28.0, Armonk, NY, USA) was used for all analyses.

## 4. Results

### 4.1. Study Characteristics

A systematic literature search was performed to identify studies that included the outcomes of interest (Figure 2). The number of identified studies were as follows: (i) population-based prevalence of endometriosis during pregnancy (*n* = 0); (ii) the diagnosis of endometriosis in women with PP during pregnancy (*n* = 0); (iii) the influence of endometriosis on the incidence of PP (*n* = 23); (iv) the influence of endometriosis on the surgical morbidity of PP patients with endometriosis (*n* = 1); and (v) molecular studies focused on PP complicated with endometriosis (*n* = 0). While endometriosis is strongly correlated with a higher prevalence of PP, our systematic research reveals that there is a scarcity of both clinical and molecular research focused on PP complicated with endometriosis.

### 4.2. Epidemiology and Outcomes

#### 4.2.1. Results of the Systematic Review

We performed a systematic literature search on the epidemiology of endometriosis during pregnancy. However, to date, no population-based studies have examined the prevalence of endometriosis during pregnancy. To discuss this topic, we performed a narrative review of the prevalence of endometriosis in the general population.

#### 4.2.2. Population-Based Prevalence of Endometriosis in the General Population

Recently, four population-based studies that examined the prevalence of endometriosis in the general population were reported [20,47,48,49,50,51]. The estimated overall prevalence of endometriosis in these population-based studies varies from 0.8–2.0% (approximately 3–5% in reproductive-aged women) [20,47,48,49,50,51]. Nevertheless, some population-based studies support the notion that the prevalence of endometriosis is increasing [51]. Additionally, since the development of ART has increased pregnancy rates among women with endometriosis [15,16,17,18,19], the number of pregnant women with endometriosis may consequently increase [19,20,21]. Therefore, future studies examining the nationwide prevalence of endometriosis during pregnancy are warranted.

### 4.3. Association between PP and Endometriosis

#### Results of the Systematic Review

We identified 20 comparator studies and three non-comparator studies that determined the influence of endometriosis on the prevalence of PP (Table 1) [7,8,11,12,25,26,27,28,29,30,31,32,33,34,35,36,37,38,39,52,53,54,55]. Of those (*n* = 23), five were population-based and 18 were retrospective studies. We found that the prevalence of PP in pregnant women with endometriosis ranged from 1.7–17.1%, while the prevalence of PP in pregnant women without endometriosis ranged from 0.5–4.3%. Among the non-comparator studies (*n* = 3), all included patients with severe endometriosis, such as women with DIE or women with rASRM stage III-IV endometriosis.

A meta-analysis was conducted to examine the influence of endometriosis on the prevalence of PP using the 20 included comparator studies (Table 1) [7,8,11,12,25,26,27,28,29,30,31,32,33,34,35,36,37,38,39,52]. A random-effect analysis was conducted due to the existence of substantial heterogeneity. In the unadjusted pooled analysis (*n* = 20), endometriosis was associated with an increased rate of PP compared to those without endometriosis (Figure 3) (OR 3.61, 95%CI 3.05–4.26; heterogeneity: *p* < 0.01, *I*^^2^^ = 73%).

Although the search terms were updated to meet the modified definition of endometriosis and the search period was extended, the studies included in the adjusted pooled analysis using the random-effects models were the same as those included in our previous meta-analysis (*n* = 12) [24]. We found that women with endometriosis were more likely to have a higher rate of PP (Figure 3) (adjusted OR 3.17, 95% CI 2.58–3.89; heterogeneity: *p* < 0.01, *I*^2^ = 86%), compared to those without endometriosis [7,12,25,26,28,31,33,34,35,37,38,39].

### 4.4. Association between Severe Endometriosis and PP

#### Results of the Systematic Review

As shown in Table 2, 11 studies determined the influence of severe endometriosis on the incidence of PP in pregnant women. Of these studies (*n* = 11), six studies compared the obstetric outcomes between women with and without DIE; one study investigated the occurrence of PP in women with and without DIE; one study evaluated the incidence of PP in women with DIE; and three studies evaluated the effect of severe endometriosis (rASRM stage III–IV).

A meta-analysis was performed to determine the impact of severe and non-severe endometriosis on the rate of PP. An unadjusted analysis using the random-effects model was performed to compare the incidence of PP in severe and non-severe endometriosis. In this analysis, women with severe endometriosis had a substantially higher rate of PP compared to those with non-severe endometriosis (Figure 4A) (*n* = 6, OR 5.22, 95% CI 2.51–10.85; heterogeneity: *p* = 0.051, *I*^^2^^ = 0%). In the unadjusted analysis using a fixed-effects model, severe endometriosis was associated with an increased prevalence of PP (Figure 4B) (OR 11.86, 95% CI 4.32–32.57; heterogeneity: *p* = 0.04, *I*^^2^^ = 55%), whereas non-severe endometriosis was not associated with an increased incidence of PP (Figure 5A) (OR 2.16, 95% CI 0.95–4.89; heterogeneity: *p* = 0.56, *I*^^2^^ = 0%).

In the sub-analysis comparison between women with non-DIE endometriosis and women without endometriosis (fixed-effects analysis), non-DIE endometriosis was not associated with an increased rate of PP (Figure 5B) (*n* = 3, OR 2.07, 95% CI 0.87–4.94; heterogeneity: *p* = 0.37, *I*^^2^^ = 0%). Similar results were observed in the comparison between women with rASRM stage I-II endometriosis versus women without endometriosis (Figure 5C) (*n* = 2, OR 1.10, 95% CI 0.08–14.49; heterogeneity: *p* = 0.16, *I*^^2^^ = 49%). The results of the meta-analysis suggest that severe endometriosis is associated with an increased rate of PP, whereas non-severe endometriosis is not.

### 4.5. Influence of Endometriosis on the Surgical Outcomes of PP Patients

In this systematic literature search, we identified one study that reported the surgical morbidity of PP patients with endometriosis [56]. This study included 24 women with PP who had posterior extrauterine adhesion. Of these patients, approximately 20% (5/24) had histologically confirmed endometriosis, 50% (12/24) were diagnosed with endometriosis based on intraoperative findings during cesarean delivery, and the remaining approximately 30% (7/24) were suspected to have endometriosis since uterine exteriorization was infeasible due to posterior extrauterine adhesion. In this study, they used propensity score-matching (PS-matching) analysis to match the patient’s background. After PS-matching (*n* = 72), median intraoperative bleeding was substantially higher in the endometriosis group (*n* = 24) compared to those in the control group (*n* = 48) (1.515 mL versus 870 mL, *p* < 0.01), whereas the transfusion rate was comparable between the two groups (2/24 (8.3%) versus 2/48 (4.2%), *p* = 0.60).

This retrospective study also examined the characteristics of PP complicated with endometriosis and found that endometriosis was correlated with a higher rate of total PP (66.7% vs. 42.7%, *p* = 0.04), posterior placenta (95.8% versus 63.2%, *p* < 0.01), longer cervical length (56.1 mm versus 36.1 mm, *p* < 0.01), and increased intraoperative bleeding (1.515 mL versus 870 mL, *p* < 0.01) compared to those without endometriosis [56]. These findings suggest that PP complicated with endometriosis appears to be a high risk factor for postpartum hemorrhage; however, the underlying cause of increased intraoperative bleeding remains unclear.

### 4.6. Diagnosis of Endometriosis in Women with PP during Pregnancy

#### 4.6.1. Results of the Systematic Review

During our systematic review, we found no studies assessing the diagnosis of endometriosis in women with PP during pregnancy. Moreover, identifying pelvic adhesions using imaging modalities remains challenging even in non-pregnant women [57]. These findings suggest that although there is a strong association between endometriosis and PP, and that identifying the presence of endometriosis is important, the diagnostic method of endometriosis, except for endometrioma during pregnancy, is understudied.

#### 4.6.2. Diagnostic Method of Extrauterine Posterior Wall Adhesion during Pregnancy

Clinicians earlier had a preconception that pregnancy had a beneficial effect on endometriosis; however, a systematic review of endometriosis during pregnancy that evaluated the progression of endometriosis during pregnancy refuted this theory [58]. The systematic review concluded that the data available on the progress of endometriosis during pregnancy showed fewer significant implications than previously reported [58]. Since no effective diagnostic method for endometriosis during pregnancy has been reported, it is necessary to determine the presence of endometriosis and/or any past surgical history of endometriosis before pregnancy through a medical interview.

Our previous retrospective study reported a magnetic resonance imaging (MRI) finding that predicts posterior uterine adhesion in women with PP. In the retrospective study, a total of 96 women with PP were included, 21 of whom had posterior uterine adhesions. However, the cause of posterior extrauterine adhesion was not clarified; thus, the study did not meet the inclusion criteria for this review. Nevertheless, we focused on and defined the angle of the uterine cervix in this study as shown in Figure 6. The mean cervical canal angle was 38.4°, and the angle decreased by 3.3° per week in women without extrauterine posterior adhesions. In contrast, the mean cervical canal angle was 5.95° in the extrauterine posterior adhesion group, which increased by roughly 0.2° per gestational week (*p* < 0.01).

## 5. Molecular Mechanisms

### 5.1. Molecular Research Focused on PP Complicated with Endometriosis

We were unable to identify any studies assessing the underlying molecular mechanisms of PP complicated with endometriosis during our systematic review. However, a few studies have reported a possible mechanism explaining the association between endometriosis and an increased rate of PP [58].

### 5.2. Role of Estrogen in Endometriosis

Previous studies have proposed a possible mechanism by which a congenital ectopic or transplanted endometrium develops into endometriosis. Furthermore, intracellular estrogen production significantly affects the etiology of endometriosis [14]. Aromatase P450 is physiologically expressed in a variety of human tissues such as adipose tissue and in the ovaries, but not usually in the endometrium, and it catalyzes the conversion of androgens to estrogens [59]. Notably, this enzyme has been found in both endometrial tissue and ectopic endometrium in patients with endometriosis [14,60]. Furthermore, in women with endometriosis, the protective activity of 17β-hydroxysteroid dehydrogenase (17β-HSD) type 2 is missing due to a loss of the enzyme [61]. This is important because 17β-HSD lowers the levels of the potent 17β-estradiol, converts it to estrone, and modulates exposure to estrogenic effects. Therefore, local estrogen production combined with the loss of protective mechanisms likely leads to high estradiol levels that characterize both endometriosis and ectopic endometrium in affected women [14].

To our knowledge, the association between local estrogen levels and the rate of PP has not been determined. However, high serum estradiol levels, when the human chorionic gonadotropin (hCG) is triggered during assisted reproductive treatment, have been associated with an increased rate of PP (OR 1.36, 95% CI 1.13–1.65) [62]. Furthermore, high serum estradiol leads to a thicker endometrium, and an endometrial thickness of >12 mm is associated with an increased risk of PP (adjusted-OR: 3.74, 95% CI: 1.90–7.34) compared to women with an endometrial thickness of <9 mm [63]. Therefore, high serum estrogen levels and a thick endometrium in women with endometriosis may lead to a high rate of PP [63]. Future studies that examine the association between serum estrogen level, local estrogen level, endometrial thickness, and the rate of PP are therefore warranted.

### 5.3. Possible Mechanisms Underlying the Increased Rate of PP in Endometriosis

Molecular studies examining the mechanisms responsible for the increased occurrence of PP patients with endometriosis are scarce. To achieve a successful human pregnancy, a number of critical stages are surpassed, including the following: (i) implantation of the blastocyst into the endometrium; (ii) successful placentation in the endometrium; and (iii) remodeling of the uterine vasculature. Furthermore, implantation deferred beyond the window of receptivity may lead to misguided implantation of the embryo, which can result in PP (abnormal placentation), PAS disorders (ectopic placentation), or placental insufficiency causing fetal growth restriction and/or a hypertensive disorder of pregnancy [58,64]. Therefore, deferred implantation is a possible mechanism underlying the increased rate of PP in pregnant women with endometriosis.

Another possible mechanism is inadequate uterine contractility. Specifically, uterine contractions are observed throughout the menstrual cycle or pregnancy in both non-pregnant and pregnant women [58]. In fact, in non-pregnant women, the uterus displays wave-like activity throughout the menstrual cycle and this activity is known as ‘endometrial waves’ [65]. These contractions seem to involve the sub-endometrial layer of the uterine myometrium [58]. Approximately 1–2 contraction waves occur per minute and last for 10–15 s in the early follicular phase with a low amplitude [66]. The frequency and amplitude of these contractions decrease during the luteal phase, likely to facilitate implantation. In the absence of blastocyst implantation, the contraction frequency remains low; however, the amplitude increases significantly, creating contractions during the menstrual period similar to those observed in labor [66].

Notably, women with endometriosis have been shown to have uterine contractions with a higher frequency, amplitude, and basal pressure tone compared to women without endometriosis [67]. Taken together, PP in women with endometriosis may result from these abnormal uterine contractions.

### 5.4. Molecular Aspects of Endometriosis-Associated Pelvic Adhesion

Previous studies have suggested that endometriosis is a disorder correlated with an inflammatory response in the peritoneal cavity and is therefore considered to be a chronic inflammatory disease [68,69]. Intraperitoneal chronic inflammation and the growth and persistence of vascularized endometrial tissue outside of the uterus are characteristic features of endometriosis [70]. In the acute phase, transplanted ectopic endometrium induces an inflammatory response, which is associated with conscription and stimulation of regulatory T (Tregs) and helper T cells [71,72]. After the acute phase, monocytes and macrophages sustain a chronic state of inflammation, which promotes angiogenesis and peritoneal adhesion formation [71].

These hypotheses are supported by studies utilizing animal models. For example, one study conducted in baboons (*Papio anubis*) examined the role of Tregs in promoting and maintaining ectopic endometrial growth [73]. The findings in this study showed that the level of Tregs decreased in the ectopic endometrium, which led to an increase in the volume of ectopic endometrium allowing for the maintenance and growth of lesions [73]. Another study examined the role of macrophages using a murine model, in which macrophages were depleted using clodronate liposomes or monoclonal antibodies [70]. This study found that tissue fragments adhered and implanted into the abdominal peritoneal wall, but ectopic endometrial cells failed to organize and develop, whereas ectopic endometrial cells were organized in control mice [70].

In terms of human data, most studies have shown an increase in the presence of inflammatory mediators (chemokines, cytokines, and prostaglandins) in the peritoneal fluid of women with endometriosis [74,75]. Moreover, few human studies have focused on the role of Tregs in women with endometriosis. For example, one case-control study including 17 histopathologically-confirmed ovarian endometriosis patients and 15 women without endometriosis compared the percentage of CD25 (high) FOXP3^+^ Treg cells in the peripheral blood and peritoneal fluid between the groups [76]. In this study, women with endometriosis were more likely to have a decreased percentage of CD25 (high) FOXP3^+^ Treg cells in the peripheral blood than women in the control group. In contrast, the percentage of these cells substantially increased in the peritoneal fluid of endometriosis patients [76]. These results suggest that intraperitoneal Treg cells can repress effector T cells and stimulate proliferation and spread of endometrial stromal cells [71].

### 5.5. Genetics: Hereditary Genetic Polymorphism

The idea of a genetic basis for the development of endometriosis stems from years of observations showing the familial occurrence of the disease [77,78,79,80]. Especially in women with monochorionic twins (*n* = 42), 40 pairs were concordant (>80%) for endometriosis. These results suggested that genetic factors are associated with the development of endometriosis.

Single nucleotide polymorphisms (SNPs) account for about 90% of the total phenotypic variability [80]. Genome-wide association studies (GWAS) can identify SNPs with effects on the risk for disease or those that affect the mean value of a quantitative trait [81] and several studies have already examined endometriosis-associated SNPs. Early GWAS on endometriosis have been reported between 2010 and 2011 [80]; two were studies on the Japanese population [82,83] and one was on the European population [84]. Since then, more than 15 GWAS have been reported and a systematic review of GWAS in endometriosis published until the end of 2019 revealed that WNT4 rs7521902, IL1A rs6542095, FN1 rs1250248, GREB1 rs13394619, and VEZT rs10859871 variants appeared to be important as the frequency was high and in terms of pathways as well as function that each gene affected in the development of endometriosis [85].

Reproducibility and validation of these variants in different populations are essential to better understand the etiology of endometriosis, optimize diagnosis, and improve the efficacy of clinical treatments [80]. Investigation of the relationship between gene expression and polymorphisms and endometriosis has the potential to contribute to developing new treatment strategies. However, to date, these studies have not yet been translated into clinical practice; thus, future molecular studies are expected to develop personalized therapies for endometriosis to decrease the rate of PP.

### 5.6. Genetics: Somatic Mutations

The association between endometriosis and ovarian cancer, especially in clear cell carcinoma or endometrioid carcinoma, led to an interest in the role of somatic mutations in genes implicated in ovarian cancer [86,87]. With regard to cancer-associated gene mutations in endometriosis, *KRAS*, *ARID1A*, *PIK3CA,* and *PPP2R1A* are frequently noted [88,89].

In non-cancer patients, an analysis of the exomes of nodules in 24 women with DIE, as compared to those in adjacent normal peritoneum was performed. Of these (*n* = 24), five women (21%) showed somatic mutations in endometriotic tissue for *ARID1A*, *KRAS*, *PIK3CA*, and *PPP2R1A* with mutations confined to glandular epithelial cells [86,90].

Exome and targeted sequencing analysis of epithelium isolated from 107 endometrioma samples compared to 82 histologically normal ectopic endometrial samples, identified hotspot mutation sites in *KRAS* and *PIK3CA* in both endometrium and endometrial epithelium [91]. The marked increase in mutant allele frequency of cancer-related genes in endometriotic epithelium suggests that retrograde flow of endometrial cells already harboring cancer-related mutations, selectively predominate in ectopic locations, leading to the development of endometriosis [91].

The impact of these mutations on the development of endometriosis is complex and highlights the need for further evaluation. Moreover, while the association between endometriosis and ovarian cancer has already been examined in numerous studies [88,92,93,94,95,96,97,98], the effect of these mutations on the prevalence of PP has not been reported. Further studies are warranted to examine the effect of these mutations on the prevalence of PP. We believe that by identifying genes undergoing mutations in endometriosis, a novel treatment that blocks endometriosis development may be able to decrease the incidence of PP.

## 6. Discussion

### 6.1. Key Findings

The key findings of this study consist of the following points: (i) severe endometriosis is related to a higher prevalence of PP, whereas non-severe endometriosis is not; (ii) only one comparator study examined the surgical morbidity of women with PP with endometriosis; (iii) there is a lack of studies assessing the underlying molecular mechanisms of endometriosis during pregnancy. Although the finding of a correlation between severe endometriosis and a higher occurrence of PP is unique, the mechanism(s) underlying the lack of an association between non-severe endometriosis and an increased rate of PP is unresolved and thus warrants future studies.

### 6.2. Comparison with Existing Literature

#### 6.2.1. Relationship between Endometriosis and PP

While previous systematic reviews, including our previous work, have already reported that endometriosis correlated to a higher occurrence of PP [9,10,58,99], it was not examined according to the severity of endometriosis; thus, we believe that the results of our systematic review are relevant. In contrast, as the findings of our systematic review were based on a univariate pooled analysis and the number of included studies was limited, we should note that this analysis may have severe bias.

#### 6.2.2. Surgical Outcomes of PP Complicated with Endometriosis

While there is a strong association between endometriosis and a higher rate of PP, data regarding surgical morbidity in PP patients with endometriosis remains scarce.

We identified one study that compared and examined the surgical outcomes between PP patients with and without endometriosis [56]. This study suggested that PP complicated with endometriosis appears to be a high risk factor for postpartum hemorrhage; however, the reason for increased intraoperative bleeding remains unclear. Although the results from this study are unique, there are still some concerns regarding the diagnosis of endometriosis [100,101,102]. For example, among the 24 patients, seven had a history of endometriosis, 12 were confirmed to have endometriosis by intraoperative findings, and the remaining cases were suspected to have endometriosis based on the infeasibility of uterine exteriorization due to extrauterine posterior wall adhesion. Therefore, it is important to note that because the quality of diagnosis for endometriosis in this study was low, the results need to be thoroughly interpreted. Moreover, the number of patients included in this study was limited; thus, future studies are needed to confirm these findings.

The hypotheses for increased intraoperative blood loss during cesarean delivery in women with PP complicated with endometriosis were tested, and the results are as follows: (i) adverse surgical outcomes were observed in women with endometriosis who had a cesarean delivery [24]; (ii) there is a possibility that endometriosis leads to increased incidence of cesarean delivery difficulty due to pelvic adhesions. Given that women with severe endometriosis, such as DIE, have been reported to have high surgical morbidity during cesarean delivery, we consider that these results support our hypotheses [8].

#### 6.2.3. Proposed Surgical Treatment for Postpartum Hemorrhage in Women with PP Complicated with Endometriosis

In general, various procedures, including, uterine compression suture, intrauterine balloon tamponade (IUBT), uterine artery embolization, and hysterectomy, can be performed in women with PP [5,103,104,105,106,107,108,109]. Of these, uterine exteriorization is crucial to perform uterine compression sutures. We consider that adhesions of the intestine and/or ovaries to the uterine posterior wall frequently make these techniques difficult to achieve.

Nevertheless, placement of the IUBT for PP complicated with endometriosis is difficult since the balloon catheter is twisted by the long, narrow cervix with strong retroflexion of the uterus [56]. Therefore, the exteriorization of the uterus is required to fix the strong retroflexion, which induces further bleeding from the uterine posterior wall due to inevitable detachment of the pelvic adhesion. Due to these factors, it is exceptionally challenging to insert the intrauterine balloon transabdominally through a uterine incision or transvaginally without performing uterine exteriorization [56].

To overcome the difficulty of IUBT placement, we propose the use of the Nelaton catheter method for cervical passage during IUBT placement. This method was originally described by Matsubara et al. as a simple and quick method for the placement of an IUBT during cesarean delivery in women with PP complicated with postpartum hemorrhage [110,111,112,113]. We modified this method into one that facilitates the insertion of IUBT without exteriorization of the uterus and allows passage through the long and narrow cervix (Figure 7). An intraoperative movie depicting this method is shown in Appendix A.

Appendix A. Intraoperative movie inserting the Bakri balloon using the modified Matsubara method.

This case involved a woman with complete PP without endometriosis, and a postpartum hemorrhage was observed during cesarean delivery. Inserting the Bakri balloon using the modified Matsubara method is a quick and easy method to place a Bakri balloon. In this case, this method successfully controlled the postpartum hemorrhage due to PP.

### 6.3. Strengths and Limitations

This study is likely to be the first to focus on the surgical outcomes of PP patients with endometriosis. We found that patients with PP and endometriosis may experience increased intraoperative blood loss during cesarean delivery. Our study also found that endometriosis is correlated with a higher occurrence of PP. Moreover, severe endometriosis is associated with an increased incidence of PP. Notably, a meta-analysis comparing severe versus non-severe endometriosis on the incidence of PP has not been conducted; thus, this can be also considered the strength of this study.

However, some notable limitations of this study exist. First, since all studies included in the review were retrospective in nature, there was an unmeasured bias in this meta-analysis. Other possible confounding factors of this meta-analysis included the various definitions of endometriosis among the identified studies, the limited number of PP patients with endometriosis, and the substandard quality of the diagnoses in some of the studies. These factors may lead to severe bias in the study. Second, we only found one similar study that determined the surgical outcomes between PP patients with and without endometriosis. However, the definition of endometriosis was ambiguous in that study, and future studies should aim to confirm our observations.

Third, women with endometriosis are more likely to conceive by ART than those without endometriosis [19,114]. Although various factors (advanced maternal age, ART pregnancy, maternal smoking, etc.) are correlated with a higher rate of PP [100,101,102], most studies [7,8,11,25,26,27,29,30,31,32,35,36,37] did not perform a multivariate analysis with adjustments for obstetric background; thus, our study cannot distinguish endometriosis as an independent risk factor for PP due to the confounding factors.

Fourth, studies that stratify the rate of PP along with the severity of endometriosis using the rASRM scoring system or DIE are limited. Importantly, a multivariate analysis with adjustments for the obstetric background has not been performed. Therefore, our analysis regarding the association between the relationship of endometriosis and the incidence of PP has the possibility of severe bias. Taken together, these are notable limitations that readers should recognize when interpreting the results of this meta-analysis.

### 6.4. Conclusions and Implications

#### 6.4.1. Implications for Practice

Endometriosis is associated with adverse maternal outcomes during cesarean delivery, such as postpartum hemorrhage, hysterectomy, and bladder injury. Furthermore, endometriosis is associated with increased intraoperative blood loss during cesarean delivery for PP, which leads to adverse surgical outcomes. Future research focused on identifying intraoperative treatments that improve the surgical outcomes of women with PP complicated with endometriosis is warranted.

#### 6.4.2. Implications for Clinical Research

Alleviating the adverse surgical outcomes of PP complicated with endometriosis as well as identifying the presence of endometriosis will help clinicians prevent the incidence of adverse events. Nevertheless, there is a scarcity of research assessing the diagnostic methods used during pregnancy. Since MRI is not routinely performed due to its high cost, future studies assessing the efficacy of diagnosing endometriosis during pregnancy using transvaginal ultrasonography are needed.

#### 6.4.3. Implications for Molecular Research

While we are moving towards unraveling the molecular and cellular pathogenesis of endometriosis, novel treatments that block the formation of pelvic adhesions and decrease the incidence of PP in women with endometriosis have not yet been developed. To that end, more research examining the mechanisms underlying the increased rate of PP in women with endometriosis is necessary for the development of these novel treatments.

## Figures and Tables

**Figure 1 biomedicines-09-01536-f001:**
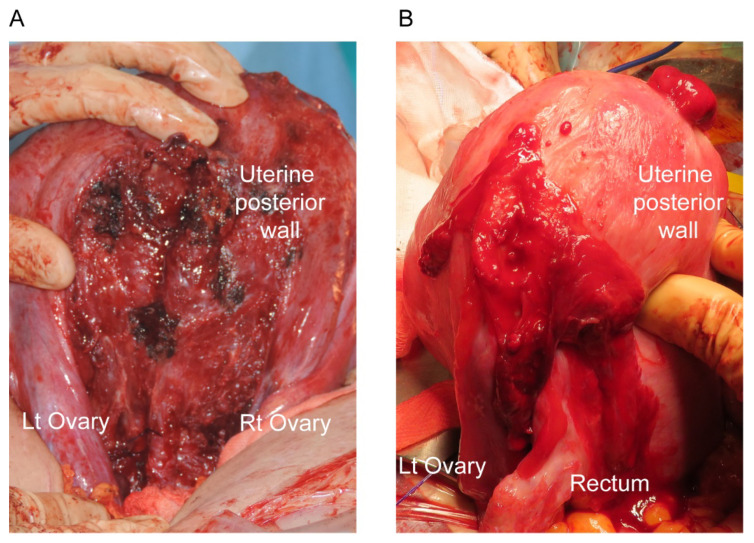
Intraoperative images during cesarean delivery of patients with endometriosis. (**A**) Intraoperative image of a patient with placenta previa (PP) and endometriosis who had a history of laparoscopic surgery for endometriosis showing the uterine posterior wall that was actively bleeding; it took a long time to control the bleeding. (**B**) Intraoperative image of a PP patient with endometriosis, but without a history of surgery for endometriosis. Strong adhesions were observed between the left posterior uterine wall and the rectum. Lt, left; Rt, right.

**Figure 2 biomedicines-09-01536-f002:**
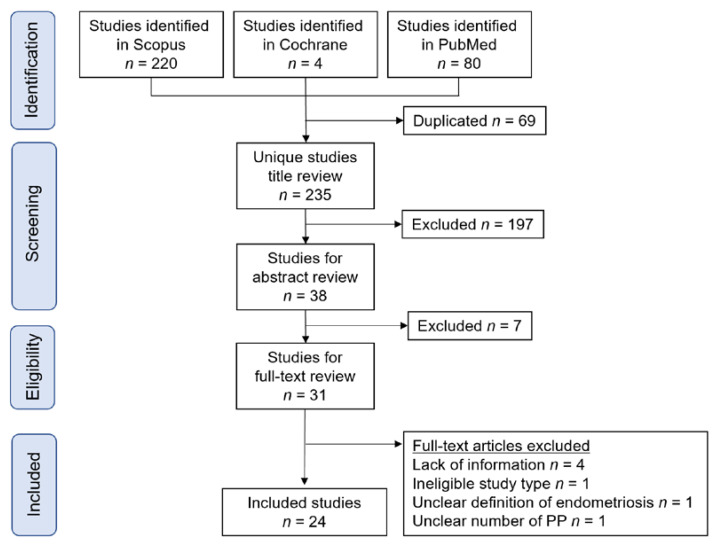
Study selection scheme of the systematic search of articles. Abbreviation: PP, placenta previa.

**Figure 3 biomedicines-09-01536-f003:**
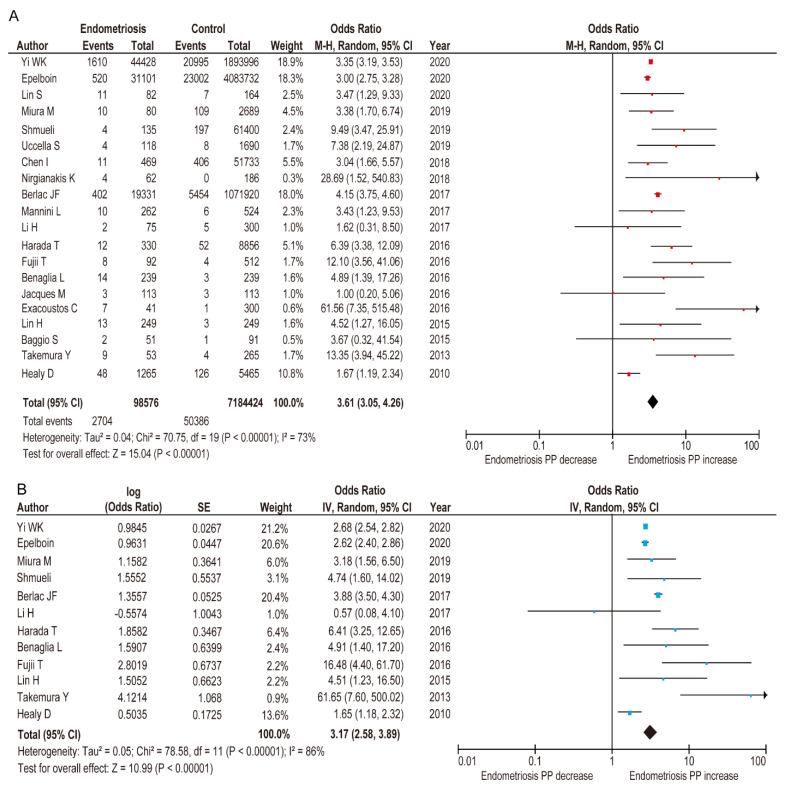
Meta-analysis of the influence of endometriosis on the occurrence of PP. Reproduced and updated, Matsuzaki S et al. [24] The association of endometriosis with placenta previa and postpartum hemorrhage: a systematic review and meta-analysis/Figure 2. Meta-analysis of the effect of endometriosis on the rate of placenta previa. Copyright (2021), with permission from Elsevier. The position of the box is a point estimate of the OR. The size of the box represents the weight of study. A horizontal line representing the 95%CI of the study result, with each end of the line representing the boundaries of the CI. Clip CIs to arrows when they exceed specified limits. The diamond represents the combined results of the studies. The pooled OR of the (**A**) unadjusted analysis for PP and (**B**) adjusted analysis for PP between women with endometriosis and women without endometriosis is shown. The forest plots were ordered within the stratum by year of publication and relative weight (%) of the study. Substantial heterogeneity was observed in A (*I*^2^ = 73%), and considerable heterogeneity was observed in B (*I*^2^ = 86%). Some values slightly vary from the original values, as calculations were performed using Revman ver. 5.4.1. Abbreviations: OR, odds ratio; SE, standard error; CI, confidence interval; PP, placenta previa.

**Figure 4 biomedicines-09-01536-f004:**
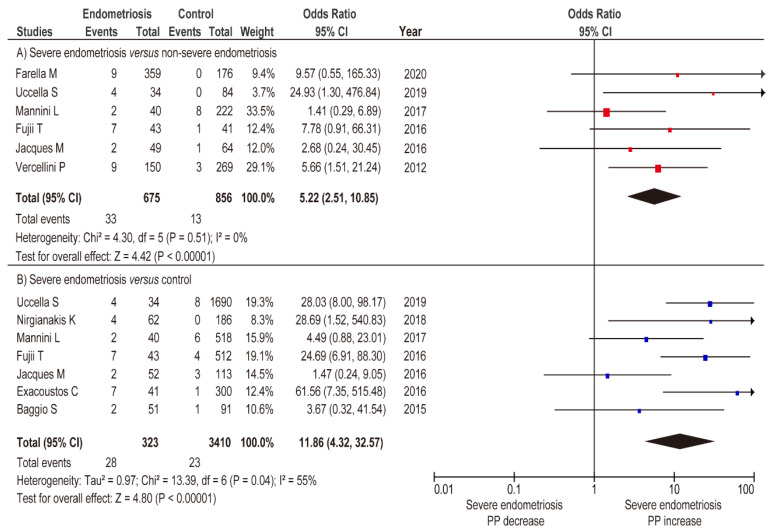
Meta-analysis of the influence of severe endometriosis on the incidence of PP. Pooled odds ratio of unadjusted analyses for PP. The position of the box is a point estimate of the OR. The size of the box represents the weight of study. A horizontal line representing the 95%CI of the study result, with each end of the line representing the boundaries of the CI. Clip CIs to arrows when they exceed specified limits. The diamond represents the combined results of the studies. (**A**) Severe endometriosis versus non-severe endometriosis, and (**B**) severe endometriosis versus control. The forest plots were arranged within the stratum by year of publication and the relative weight (%) of the study. No heterogeneity was observed in the analysis of A (*I*^2^ = 0%), while moderate heterogeneity was observed in the analysis of B (*I*^2^ = 55%). Therefore, a fixed-effects analysis was applied to panel A, whereas a random-effect analysis was used for panel B. Some values may be slightly different from the original values, as calculations were performed using Revman ver. 5.4.1. Abbreviations: OR, odds ratio; SE, standard error; CI, confidence interval; PP, placenta previa.

**Figure 5 biomedicines-09-01536-f005:**
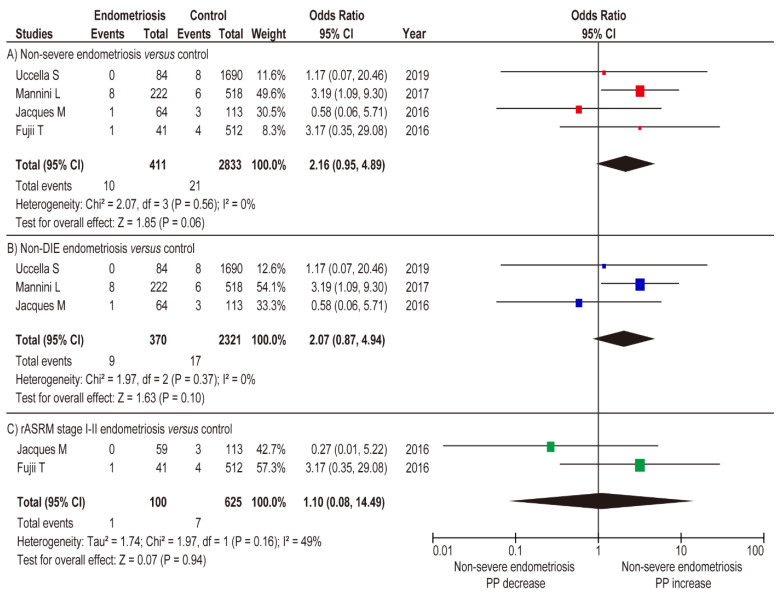
Influence of non-severe endometriosis on the occurrence of PP. Pooled odds ratio of unadjusted analyses for PP. The position of the box is a point estimate of the OR. The size of the box represents the weight of study. A horizontal line representing the 95%CI of the study result, with each end of the line representing the boundaries of the CI. The diamond represents the combined results of the studies. (**A**) non-severe endometriosis versus control, (**B**) non-DIE versus control, and (**C**) rASRM stage I–II endometriosis versus control. The forest plots were arranged within the stratum by year of publication and the relative weight (%) of the study. Low heterogeneity was observed in the analysis of panel C (*I*^2^ = 49%), whereas no heterogeneity was observed in the analysis of panels A and B (*I*^2^ = 0%). Therefore, a fixed-effects analysis was applied to panels A and B, whereas a random-effect analysis was used for panel C. Some values may slightly vary from the original values, as calculations were performed using Revman ver. 5.4.1. Abbreviations: OR, odds ratio; SE, standard error; CI, confidence interval; and PP, placenta previa.

**Figure 6 biomedicines-09-01536-f006:**
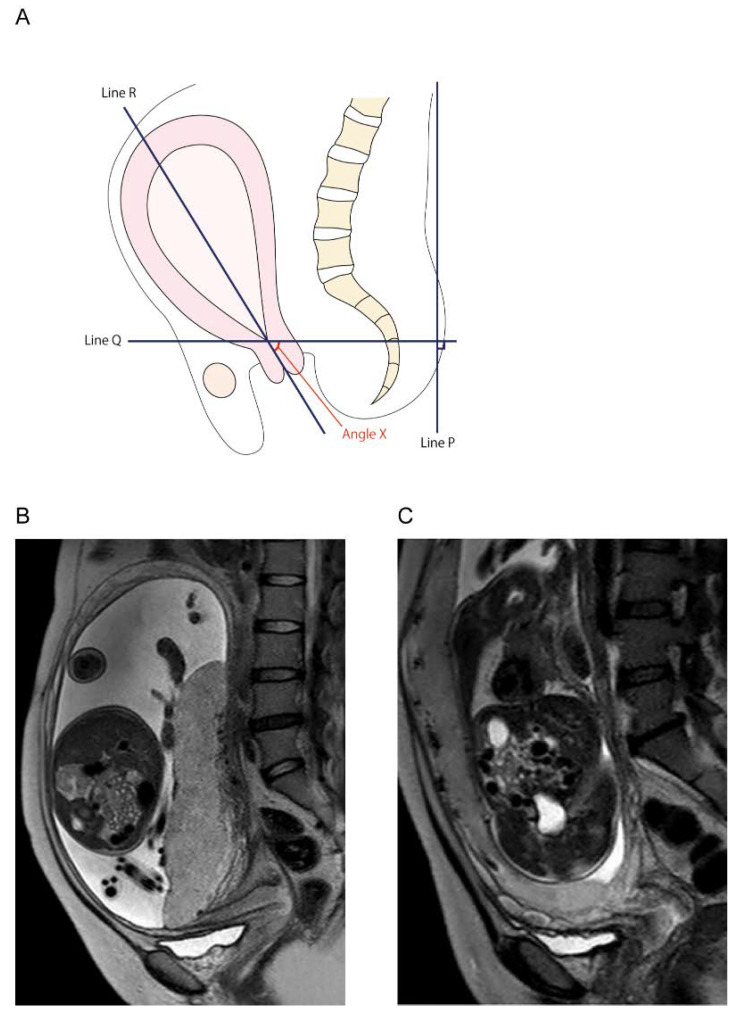
Determination of the cervical canal angle. Modified from BMC Surg. 6 January 2021;21:10. Nagase Y et al. Placenta previa with posterior extrauterine adhesion: clinical features and management practice/Supplemental Figure S1, Determination of the horizontal cervix sign [56]. (**A**) The method for measuring the angle of the cervical canal on magnetic resonance images. Line P is a straight line through a broad part of the patient’s back. Line Q is a line perpendicular to line P. Line R is a line passing through the internal os to the external os, and Angle X is the angle formed by lines Q and R (defined as the cervical canal angle). (**B**) Typical magnetic resonance imaging of a woman with PP complicated with endometriosis. (**C**) A typical magnetic resonance imaging of a PP patient without endometriosis. Abbreviation: PP, placenta previa.

**Figure 7 biomedicines-09-01536-f007:**
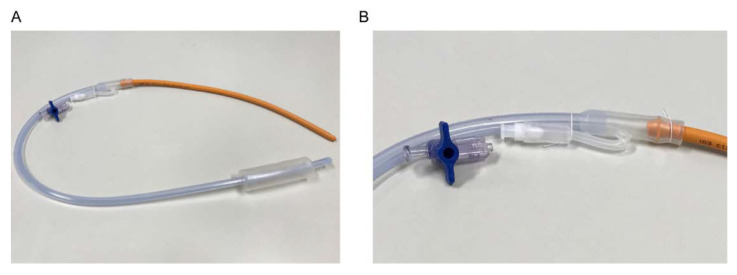
Preparation of the modified Matsubara Nelaton catheter. Reproduced from Taiwan J Obstet Gynecol. 2019;58:721-722. Matsuzaki S et al. Letter to “Cervical varices unrelated to placenta previa as an unusual cause of antepartum hemorrhage: A case report and literature review.” Successful management of postpartum hemorrhage due to cervical varix: modified Matsubara-Nelaton method using Bakri balloon/ Figure 1, Transvaginal sonography and speculum examination results., Copyright (2019) with permission from Elsevier [111]. (**A**,**B**) A Nelaton rubber tube is connected to perform the modified Matsubara Nelaton method as previously described [110,111,112,113]. However, we modified the Nelaton rubber tube type and its connections used from that of the original method. A No. 13 Nelaton rubber tube was attached to the blood drainage port of the Bakri balloon.

**Table 1 biomedicines-09-01536-t001:** Influence of endometriosis on the incidence of placenta previa (PP).

Author	Year	Area	No.	Endo No.	Severe Cases ^#^	Study Type	IVF		Previa	
							Endo (%)	Control (%)	Endo	Control
Comparator studies: women with endometriosis versus women without endometriosis										
Epelboin, S. [25]	2020	FRN	4114833	31101	--	Nationwide	18.2%	0 *	1.7%	0.6%
Yi, W.K. [26]	2020	KOR	1938424	44428	--	Nationwide	--	--	3.6%	1.1%
Lin, S. [27]	2020	CHN	246	82	--	Retro	--	--	13.4%	4.3%
Shumueli, A. [28]	2019	ISR	61535	135	--	Retro	19.3%	2.6%	3.0%	0.3%
Miura, M. [12]	2019	JPN	2769	80	--	Retro	28.7%	12.8%	12.5%	4.1%
Uccella, S. [11]	2019	ITA	1808	118	DIE	Retro	--	--	3.4%	0.5%
Chen, I. [29]	2018	CAN	52202	469		Nationwide	--	--	2.4%	0.8%
Nirgianakis, K. [30]	2018	CHE	248	62	DIE	Retro	24.2%	27.4%	6.5%	0.0%
Berlac, J.F. [7]	2017	DEN	1091251	19331	--	Nationwide	19.0%	3.3%	2.1%	0.5%
Li, H. [31]	2017	CHN	375	75	--	Retro	--	--	2.7%	1.7%
Mannini, L. [32]	2017	ITA	786	262	DIE	Retro	26.0%	10.1%	3.8%	1.1%
Benaglia, L. [33]	2016	ITA	478	239	--	Retro	All	All	5.9%	1.3%
Jacques, M. [52]	2016	FRN	226	113	DIE	Retro	All	All	2.7%	2.7%
Fujii, T. [34]	2016	JPN	604	92	rASRM	Retro	All	All	8.7%	0.8%
Exacoustos, C. [8]	2016	ITA	341	41	DIE	Retro	--	--	17.1%	0.3%
Harada, T. [35]	2016	JPN	9186	330	--	Nationwide	8.8%	2.2%	3.6%	0.6%
Baggio, S. [36]	2015	ITA	144	51	DIE	Retro	--	--	3.9%	1.1%
Lin, H. [37]	2015	CHN	498	249	--	Retro	0	0	5.2%	1.2%
Takemura, Y. [38]	2013	JPN	318	53	--	Retro	All	All	17.0%	1.5%
Healy, D. [39]	2010	AUS	6730	1265	--	Retro	All	All	3.8%	2.3%
Non-comparator studies: women with endometriosis										
Tuominen, A. [53]	2021	FIN	243	243	DIE	Retro	--	--	11.1%	--
Farella, M. [54]	2020	FRN	535	535	rASRM	Retro	--	--	1.7%	--
Vercellini, P. [55]	2012	ITA	419	419	DIE	Retro	0	--	2.9%	--

Reproduced and updated the data from Am J Obstet Gynecol MFM. 2021;3:100417. Matsuzaki S et al. [24].The association of endometriosis with placenta previa and postpartum hemorrhage: a systematic review and meta-analysis/Supplemental Table S2, Results of a systematic review of the effect of endometriosis on the prevalence of placenta previa and postpartum hemorrhage. Copyright (2021), with permission from Elsevier. The number (percentage per column) is presented. * IVF pregnancies without endometriosis were excluded from the analysis. ^#^ We defined severe cases as women with DIE or rASRM stage III-IV endometriosis. Abbreviations: --, not applicable; Endo, endometriosis; No., number of included cases; retro, retrospective; AUS, Australia; JPN, Japan; CHN, China; ITA, Italy; DEN, Denmark: CHE, Switzerland; CAN, Canada; KOR, Korea; and ISR, Israel; DIE, deep infiltrating endometriosis; rASRM; revised American Society for Reproductive Medicine.

**Table 2 biomedicines-09-01536-t002:** Effect of severe and non-severe endometriosis on the prevalence of PP.

Author	Year	No.	Endo No.	Severe Endo ^#^	Severe No.	Non-Severe No.	Control No.	Previa		
								Severe	Non-Severe	Control
Comparator studies: women with DIE versus women without endometriosis										
Uccella, S. [11]	2019	1808	118	DIE	34	84	1690	4/34 (11.8%)	0/84 (0%)	8/1690 (0.5%)
Nirgianakis, K. [30]	2018	248	62	DIE	62	--	186	4/62 (6.5%)	--	0
Mannini, L. [32]	2017	786	262	DIE	40	222	518	2/40 (5.0%)	8/222 (3.6%)	6/518 (1.2%)
Jacques, M. [52] ^†^	2016	226	113	DIE	49	64	113	2/49 (4.1%)	1/64 (1.6%)	3/113 (2.7%)
Exacoustos, C. [8]	2016	341	41	DIE	41	--	300	7/41 (17.1%)	--	1/300 (0.3%)
Baggio, S. [36]	2015	144	51	DIE	51	--	93	2/51 (3.9%)	--	1/93 (1.1%)
Non-comparator studies: women with DIE										
Tuominen, A. [53]	2021	243	243	DIE	243	--	--	27/243 (11.1%)	--	--
Vercellini, P. [55]	2012	419	419	DIE	150	269	--	9/150 (6.0%)	3/269 (1.1%)	--
Comparator studies: endometriosis was classified with rASRM										
Farella, M. [54]	2020	535	535	rASRM	359	176	--	9/359 (2.5%)	0	--
Fujii, T. [34]	2016	604	92	rASRM	43	41	512	7/43 (16.3%)	1/41 (2.4%)	4/512 (0.8%)
Jacques, M. [52] ^†^	2016	226	113	rASRM	52	59	113	2/52 (3.8%)	0	3/113 (2.7%)

The number (percentage per column) is presented. ^#^ We defined severe cases as women with DIE or rASRM stage III–IV endometriosis. ^†^ In this study, severe endometriosis was determined both in the classifications of DIE and rASRM stage III-IV. Some cases overlapped, and some were examined as women with severe endometriosis. Abbreviations: Endo, endometriosis; No., number of included cases; DIE, deep infiltrating endometriosis; rASRM; revised American Society for Reproductive Medicine.

## Data Availability

All the studies included in this study are published in the literature.

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
