# Peer review of "Placenta Previa Complicated with Endometriosis: Contemporary Clinical Management, Molecular Mechanisms, and Future Research Opportunities"

_biomedicines, 2021, doi:10.3390/biomedicines9111536_

Round 1

Reviewer 1 Report

     Meta-research by Matsuzaki S, Nagase Y, Ueda Y, et al. "Placenta previa complicated with endometriosis: Contemporary clinical management, molecular mechanisms, and future research opportunities" addresses several important issues about the relationship between endometriosis and placenta previa (PP). Endometriosis (EM) is a prevalent (5 to 15% in reproductive-aged women) chronic inflammatory disease, so studies in this area are of great value.

     Reviewed are 24 research papers meeting the author’s inclusion criteria. The study selection procedure (24 of 235 articles found in literature) looks reasonable, statistical methods (chi-square test and Fisher’s exact test) are correct for such an analysis. Meta-analysis of these studies reveals a strong, statistically, and clinically significant link between EM and PP, as well as poor surgical outcomes during cesarean delivery. The authors also found that severe endometriosis was significantly associated, with an increased prevalence of PP, whereas non-severe endometriosis was not.

     The authors suggest some interesting hypotheses that can explain the existing correlations. For example, they suggest that deferred implantation can be a mechanism underlying the increased rate of PP in pregnant women with endometriosis. Also, the interesting idea is that, in the acute phase, transplanted ectopic endometrium induces an inflammatory response, which is associated with conscription and stimulation of regulatory and helper T cells and, after the acute phase, monocytes and macrophages sustain a chronic state of inflammation.

     This supposition is in line with the broadly discussed hypothesis that endometriosis is a disorder correlated with an inflammatory response. To this end, the recently established nature of endometriosis as a genetically determined disease could bring a new dimension to such discussions. It would be interesting to know the author’s opinion, but the authors regrettably do not discuss the genetic approach at all. With known genes responsible for endometriosis, a novel treatment that blocks endometriosis development may be able to decrease the incidence of PP. 

     On a more practical note, the presented meta-analysis outcomes can have important implications for the clinical practice. 

     To the drawbacks, I would refer certain reuse of the data which authors have already reviewed in their early publications, e.g., Matsuzaki S, Nagase Y, Ueda Y, et al., The association of endometriosis with placenta previa and postpartum hemorrhage: a systematic review and meta-analysis. Am J Obstet Gynecol MFM. 2021 Sep;3(5):100417. doi: 10.1016/j.ajogmf.2021.100417. 

     Finally, the manuscript falls within the aim of the Journal, the topic is challenging and attracts the readers’ attention. Some drawbacks should not be the obstacles to publication, so the article is recommended for publication in the Biomedicines journal.

Author Response

Editor and Reviewers’ comments

We would like to thank the Editor and the Reviewers for the helpful comments. The following are our point-by-point responses to the comments and explanations regarding the revisions made to the manuscript. The line numbers of the revised text are indicated. The revisions made in the manuscript are indicated using the “track changes” function of Microsoft Word.

Reviewer #1

Meta-research by Matsuzaki S, Nagase Y, Ueda Y, et al. "Placenta previa complicated with endometriosis: Contemporary clinical management, molecular mechanisms, and future research opportunities" addresses several important issues about the relationship between endometriosis and placenta previa (PP). Endometriosis (EM) is a prevalent (5 to 15% in reproductive-aged women) chronic inflammatory disease, so studies in this area are of great value.

     Reviewed are 24 research papers meeting the author’s inclusion criteria. The study selection procedure (24 of 235 articles found in literature) looks reasonable, statistical methods (chi-square test and Fisher’s exact test) are correct for such an analysis. Meta-analysis of these studies reveals a strong, statistically, and clinically significant link between EM and PP, as well as poor surgical outcomes during cesarean delivery. The authors also found that severe endometriosis was significantly associated, with an increased prevalence of PP, whereas non-severe endometriosis was not.

     The authors suggest some interesting hypotheses that can explain the existing correlations. For example, they suggest that deferred implantation can be a mechanism underlying the increased rate of PP in pregnant women with endometriosis. Also, the interesting idea is that, in the acute phase, transplanted ectopic endometrium induces an inflammatory response, which is associated with conscription and stimulation of regulatory and helper T cells and, after the acute phase, monocytes and macrophages sustain a chronic state of inflammation.

Reply:

We appreciate these useful comments from the Reviewer. We have revised the manuscript carefully according to the Reviewer’s comments. We believe the revised manuscript now addresses the Reviewer’s concerns.

Reviewer #1, Comment 1

This supposition is in line with the broadly discussed hypothesis that endometriosis is a disorder correlated with an inflammatory response. To this end, the recently established nature of endometriosis as a genetically determined disease could bring a new dimension to such discussions. It would be interesting to know the author’s opinion, but the authors regrettably do not discuss the genetic approach at all. With known genes responsible for endometriosis, a novel treatment that blocks endometriosis development may be able to decrease the incidence of PP.

 On a more practical note, the presented meta-analysis outcomes can have important implications for the clinical practice.

Reply: Lines 465 to lines 514

Thank you for your helpful comments. As the Reviewer has pointed out, we had earlier not discussed the genetic approach. We have added a brief overview of genetic studies on endometriosis. We have added the sections titled “5.5 Genetics: Hereditary Genetic Polymorphism” and “5.6. Genetics: Somatic Mutations” as per the Reviewer’s suggestion.

Reviewer #1, Comment 2

To the drawbacks, I would refer certain reuse of the data which authors have already reviewed in their early publications, e.g., Matsuzaki S, Nagase Y, Ueda Y, et al., The association of endometriosis with placenta previa and postpartum hemorrhage: a systematic review and meta-analysis. Am J Obstet Gynecol MFM. 2021 Sep;3(5):100417. doi: 10.1016/j.ajogmf.2021.100417.

Reply: Table 1 (lines 220 to 224), Figure 3 (lines 246 to 249)

Thank you for your valuable comments regarding the re-used data of our prior publication. According to the Reviewer’s suggestion, we have cited our earlier publication and obtained copyright permission to reuse the data. We have clarified regarding the copyright permission in the legends in Table 1 and Figure 3.

Finally, the manuscript falls within the aim of the Journal, the topic is challenging and attracts the readers’ attention. Some drawbacks should not be the obstacles to publication, so the article is recommended for publication in the Biomedicines journal.

Reply:

Thank you for the positive comments on our manuscript. We believe that the revised manuscript should address the concerns of the Reviewer.

Reviewer 2 Report

In this meta-analysis, Matsuzaki et al. show that endometriosis classified as severe was associated with increased risk of placenta previa, while non-severe (or no) endometriosis was not significantly associated with placenta previa. Because their literature search found no studies on a mechanistic relationship between these two conditions, the authors call for more research into how endometriosis and placenta previa might be causally related. Overall, this is a very well written and useful study. I only have a few comments:

-- Section 6.2.2 of the Discussion unnecessarily contains a restatement of some of the Results, including sample sizes and statistics.

-- In section 6.1, the second “Key finding” of the current study (increased blood loss in PP patients with vs. without endometriosis) did not come from this meta-analysis, but was reported in a single cited study. Though the authors are clear about its origin, they should ideally rephrase this and other instances where they state they have “found” this already published result. 

Author Response

Editor and Reviewers’ comments

We would like to thank the Editor and the Reviewers for the helpful comments. The following are our point-by-point responses to the comments and explanations regarding the revisions made to the manuscript. The line numbers of the revised text are indicated. The revisions made in the manuscript are indicated using the “track changes” function of Microsoft Word.

Reviewer #2

In this meta-analysis, Matsuzaki et al. show that endometriosis classified as severe was associated with increased risk of placenta previa, while non-severe (or no) endometriosis was not significantly associated with placenta previa. Because their literature search found no studies on a mechanistic relationship between these two conditions, the authors call for more research into how endometriosis and placenta previa might be causally related. Overall, this is a very well written and useful study. I only have a few comments:

Reply:

Thank you for your positive comments. We have revised the manuscript according to your valuable comments. Please refer to our reply for each comment.

Reviewer #2, Comment 1

-- Section 6.2.2 of the Discussion unnecessarily contains a restatement of some of the Results, including sample sizes and statistics.

Reply: Lines 539 to 545

We appreciate these valuable comments from the Reviewer. We have deleted the restatement of some of the results of the cited study.

-- In section 6.1, the second “Key finding” of the current study (increased blood loss in PP patients with vs. without endometriosis) did not come from this meta-analysis, but was reported in a single cited study. Though the authors are clear about its origin, they should ideally rephrase this and other instances where they state they have “found” this already published result.

Reply: lines 519 to 521

Thank you for your helpful comments. As the Reviewer has pointed out, the results were not obtained from the results of this meta-analysis and were obtained from a single cited study. According to the Reviewer’s comments, we have revised the second key finding of the current study.